# The Influence of Cryogenic Conditions on the Process of AA2519 Aluminum Alloy Cracking

**DOI:** 10.3390/ma13071555

**Published:** 2020-03-27

**Authors:** M. Kotyk, D. Boroński, P. Maćkowiak

**Affiliations:** Faculty of Mechanical Engineering, University of Science and Technology in Bydgoszcz, 85-796 Bydgoszcz, Poland; dariusz.boronski@utp.edu.pl (D.B.); pawel.mackowiak@utp.edu.pl (P.M.)

**Keywords:** fracture mechanics, cryogenic conditions, lightweight materials, aluminum alloy

## Abstract

This study presents the results of tests involving determining quantities used to describe fracture toughness of a heat-treated AA2519 aluminum alloy applied in, among other things, constructing American military amphibians. These quantities were determined using the J–R curve method for two temperature values, 293 K and 77 K. The low temperature was provided by putting the tested specimen into a liquid nitrogen bath and keeping it there throughout the experiment. Based on the tests results, cryogenic conditions cause an increase in the maximum experimental value of the J–J_Q_ integral, from 66.3 to 87.3 kJ/m^2^ Moreover, an analysis of the fatigue fracture microstructure revealed differences between specimens tested in ambient temperature and those tested in cryogenic temperature.

## 1. Introduction

Due to their lower density, aluminum alloys are often used as an alternative to steel in constructing different structures. The research results also contribute to improvements in the mechanical properties of the described materials [1]. The development of new manufacturing technologies and heat treatments were real breakthroughs in using aluminum alloys, including durals in the shipbuilding, military and aviation industries. Designers develop these materials not only because they are light while maintaining good mechanical properties, compared to steel, but also due to their special features including small inner resistance, high plasticity or particularly desired corrosion resistance [2].

Durals belong to these materials, and AA2519 aluminum alloy is a dural that is considered to be particularly interesting. NASA and the American army have been interested in it since the 1980s [3,4] and currently, a new application has been found for it.

Recently AA2519 has become particularly interesting due to the emergence of a completely new explosion-welded Al–Ti material in which AA2519 aluminum alloy is one of the constituent materials. Consequently, the existence of such a composite has focused the attention of researchers who study its mechanical properties, including static properties, fatigue life and fracture toughness [5,6,7,8]. A completely new application provokes research on this alloy in a new context. However, this does not mean that this material has not been previously recognized in the literature on the subject.

AA2519 is commonly used in ships. The literature provides information on its application in American military amphibians. The interest in the material results from its high corrosion resistance, viewed as insusceptibility to external factors including strongly saline seawater [9].

This alloy is also commonly used in aviation as a material for aircraft sheathing. Lin et al. dealt with the application of AA2519 in aircraft sheathing type to reduce mass without losing mechanical properties [10].

A number of articles have been found on the subject of heat treating AA2519 aluminum alloy and its microstructure shaping [11,12,13]. There was also an article about numerically modeling the material cracking process [14]. However, few studies of the alloy’s mechanical properties have been found.

One is the work of Zuiko et al. [15], which includes the results of testing the material static resistance in terms of heat unification. Their test results confirmed the positive influence of heat treatment on AA2519.

Selected characteristics of AA2519 were also described by Plonka et al. [16], where the influence of successive stages of rolling on the final strength of the material was determined. Aluminum that was 10 mm wide was rolled down to 3 mm and, after the last rolling, its static strength increased from 254 to 374 MPa with a simultaneous increase of toughness expressed in the Brinell scale.

AA2519 has also been tested for fracture toughness. Hynes and Gangloff [2] carried out tests to determine the fracture toughness in terms of elasticity–plasticity. They also modeled brittle fracture toughness based on their test results. Their study presents the influence of different alloy additives and higher temperature on the fracture toughness.

Tests of a similar material, that is AA2519, for application in the aerospace industry have been carried out by NASA. In the final report [17], there are results for a series of mechanical property tests on AA2159, including its resistance to elastic–plastic fracture in cryogenic conditions. In conclusion, it stated that the described conditions cause improvement in the material’s mechanical properties. Our research team also performed tests involving determining the basic mechanical properties for AA2519 under ambient (293 K) and cryogenic conditions (77 K), which are described in the works of [5,6,7,8,18,19,20]. In this article, however, apart from mechanical characteristics describing the material fracture toughness, there are also differences in the specimen fracture appearance obtained for both temperature conditions.

Application of the Al-Ti material in the aviation, shipbuilding and aerospace industries makes it necessary to do research on both the composite and base materials in new, extreme conditions, including those relating to temperature. The literature review has not provided works with the results of tests involving determining the material fracture toughness in cryogenic conditions.

Studies on the influence of cryogenic conditions on mechanical characteristics and technological processes have already been widely conducted. Some of them concern the improvement of material machining technology. An interesting example of such research is presented in the article [21], where the results of studies on the influence of using liquid nitrogen for cooling of AISI 304 material during rolling are presented. In the case of research on the use of liquid nitrogen for machining, the main problem is the application of the liquid nitrogen stream to the workpiece. This is different when determining the mechanical characteristics of materials under cryogenic conditions. To ensure cryogenic conditions throughout the experiment, a previously prepared climate chamber should be used.

An example of using the cryogenic chamber to determine mechanical properties of materials at the cryogenic temperature of joints is the one in the article [22]. Graphene [23] and alloy steels [24] and austenitic steels [25] are also tested under cryogenic conditions.

Similar as in the case of metals, mechanical properties using the climatic chamber are also determined for composite materials [26] and polymeric materials [27].

No articles addressing the issues presented in this article have been found in the literature.

## 2. Tested Material

### Basic Characteristics and Application

AA2519 aluminum alloy, whose chemical composition is presented in Table 1, is an AlCuMgMn group alloy. Their distinctive feature is high mechanical strength while maintaining plasticity, referred to as a large interval between R_p0.2_ and strain corresponding to stress, with an immediate tensile strength. This feature is particularly useful in military objects such as multilayer ballistic shields. Ultra-strong and hard materials need to be placed on the external side of the shield to destroy missiles while plastic materials (inside the shield) absorb the impact energy. In simple terms, this feature can be referred to as ballistic resistance. AA2519 exhibits this property [28].

Shortly after the material was developed, its application was abandoned. Once a technology for its plastic treatment was provided, it became popular and available in forms other than sheets and plates [9,29,30].

Raw sheets were cut into strips and subjected to heat treatment that involved saturating an aluminum compound with copper to create a θ phase and accelerated ageing to provide constant mechanical properties during operation. From a technological point of view, heat treatment involved heating the material to 530–550 °C for two hours, cooling the material in water and accelerated ageing for 10 h at 165 °C.

## 3. Testing Method and Test Stand

### 3.1. Testing Method

The tests of the impact of cryogenic conditions on the maximum experimental value of the J–J_Q_ integral were conducted using an Instron 8501 hydraulic strength testing machine. The analyzed value was determined for compact specimens C(T). They were cut from a sheet by electro-erosion methods. After being cut, the specimens were mounted on the strength testing machine to generate an 8 mm long fatigue fracture. It needs to be highlighted that the fractures of all the specimens were generated under ambient conditions (in the air). Basic dimensions of the specimens used for tests are presented in Figure 1. It should be noted that all specimens were cut in one direction, consistently with the sheet rolling direction. In effect, the cracking surface was perpendicular to the direction of rolling. A few experiments were conducted for each test temperature; however, the study provides averaged results from four selected tests. It also needs to be noted that, from the point of view of the test procedure, the final sample thickness, 10 mm, may be too small to meet all the conditions to consider the maximum experimental value of the J integral as a material mechanical characteristic. However, this thickness was selected because it is planned for a comparison between experimental values characterizing the fracture toughness of the AA2519 alloy with the same values determined for layered materials: AA2519–AA1050–Ti6Al4V. The above-mentioned sheet thickness, 10 mm, is the only available thickness of the Al–Ti layered material. For this reason, the maximum experimental value J–J_Q_ will be used later in this article.

As mentioned before, due to difficulties in observing the surface of a specimen bathed in liquid nitrogen, fatigue cracks were generated at ambient temperature. Thus, the specimen was loaded using a zero-pulsating cycle in which the maximum value of the applied force was 5 kN, at a 5 Hz frequency. Loading cycles were repeated until the fracture length was 18 mm measured on the specimen side from the force application point to the top of the fatigue fracture. This quantity includes the length of a mechanical notch.

The determination of the maximal experimental value of the J–J_Q_ integral was performed in two steps. The main control signal was machine piston displacement. The first load reduction of the specimen was started with the displacement value equal to the successive cycles, that is, 0.05 mm. Then, the load was reduced by 15% of the force value recorded at the cycle maximum point. After load reduction, the specimen was loaded again, with displacement control using a machine with a 0.05 mm step. It should be noted that the piston travel speed was 0.02 mm/s. As in the previous case, the experiment was continued until a plastic fracture was created, which meant the destruction of the specimen.

An original program based on digital image correlation was used for full automation of the cracking process. The image recorded by video cameras placed on both sides of the sample was analyzed by the program algorithms. After the expected fracture length was achieved, the program gave a command to stop the machine and the fatigue fracture was not generated any more. A scheme of the software operation is demonstrated in Figure 2. The crack tip is overshadowed by a graphical representation of the motion gradients visible to the user as an area whose displacement value corresponds to one of 255 colors.

Obviously, the fracture toughness tests were preceded by a series of experiments that provided data on the tensile properties of the AA2519 alloy. Details regarding these experiments can be found in Boroński et al. [18].

### 3.2. Cryogenic Conditions

As mentioned before, some of the tests were conducted under cryogenic conditions (77 K). These tests involved bathing the tested specimen in liquid nitrogen. To provide a constant temperature throughout the experiment, the specimen was put into an environmental chamber whose bottom pivot was fixed directly to the strength testing machine grip. Apart from heat insulation elements, the chamber was made from Inconel 625, while the cover was made from Teflon to avoid freezing. The test stand is shown in Figure 3.

It is difficult to define the exact amount of liquid nitrogen to be used for tests. It is natural, though, that the amount of nitrogen required to cool the chamber needs to be bigger than its capacity since, at the beginning of the tests, the chamber temperature was essentially the temperature of the environment. After cooling the chamber, the amount of liquid nitrogen required for determining individual values of the integral J, J_i_ and ultimately the maximum experimental J_Q_ value. significantly decreased. However, specimen replacement let in a significant amount of heat. Therefore, it was necessary to provide liquid nitrogen in an amount exceeding the chamber volume by a few times to ensure its constant supply.

### 3.3. Metallurgic Tests

A detailed analysis was carried out to identify the impact of cryogenic conditions on the characteristics of the tested alloy fracture toughness. In order to explain the reasons for the differences in the analyzed mechanical characteristics of the aluminum alloy under ambient and cryogenic conditions, it was decided to compare the microstructure of breakthroughs of the samples tested at both described temperatures. In order to do so, using the scanning electron microscope, photographs of the breakthrough surfaces of the samples tested earlier were taken. The device worked at 15 kV voltage. The analysis was performed for magnitudes of ×500 and ×1000, depending on the needs. It should be stressed that, prior to the tests, the specimens were purified in an ultrasonic cleaner. No other treatment was applied to the specimen prior to observation. The analyzed surface was not prepared for tests in any other way. The specimen was scanned from the fatigue fracture head to the specimen top. Details on the results of the sample breakthrough analysis can be found further down in this article.

## 4. Tests Results and Analysis

### 4.1. Mechanical Characteristics

Figure 4 show the curves obtained for each sample in the process of determining the maximum experimental value, J_Q_ under ambient conditions. Figure 5 shows the same curves, i.e. load-COD, but determined during cryogenic testing. The curves represent dependencies between the applied force and the specimen opening for each cycle of the loading–fracture opening. It must be noted that, due to the displacement of the force application point in relation to the opening measurement, the provided experimental values were adequately corrected (calculated) based on a model described by Kotyk and Boroński [31].

Under cryogenic conditions, higher force values were recorded during subsequent cycles. Moreover, the shape of the load–crack opening displacement (COD) line was more orderly. As shown in the article [18] the aluminum alloy in cryogenic conditions is characterized by higher stiffness, understood as a higher Young’s modulus value. The reason for this is the reduction of voids between atoms of the material. This makes the material more compact and, as a result, with the same load value, it reduces the size of the plastic areas around the apex of the crack. Larger plastic areas when tested under ambient conditions cause the material to deform in a larger volume, which results in a disruption of the regular characteristic of the obtained load–COD. Under cryogenic conditions, when plastic areas are smaller, the curve is more regular because plastic deformation occurs in a smaller volume.

### 4.2. Experimental Value of J–J_Q_ Integral

While determining the fracture toughness of materials, it is necessary to use specimens with a maximum thickness to provide an appropriate plastic zone size. Since the experimental values obtained for AA2519 are a point of reference for further comparative tests, it was decided to use 10 mm thick specimens. The above-mentioned thickness was not chosen randomly because the layered material A2519–AA1050–Ti6Al4V, production of which requires the application of the analyzed aluminum alloy, had a thickness equal of 10 mm. It needs to be noted that the ASTM E 1820 standard allows using specimens with such thickness; however, further into the study, the authors use the concept of the maximum experimental value of the J–J_Q_ integral, which should not be identified with fracture toughness J_IC_.

The presented results were obtained using the earlier described compact tension specimens with similar fatigue fracture length. The R curve method was used to determine the maximum experimental value of the J–J_Q_ integral.

### 4.3. Tests Results

The J integral values were calculated using the information included in ASTM E 1820-18 and according to mathematical dependence:(1)$$J=J_{pl}+J_{el}$$
where:(2)$$J_{pl}=\frac{\eta_{pl}\cdot A_{pl}}{B_{n}\cdot b}$$
(3)$$J_{el}=\frac{K^{2}\left(1-\upsilon^{2}\right)}{E}$$
and
(4)$$K=\frac{P}{BB_{N}W}\left[\frac{\left\{\left(2+\frac{a}{W}\right)\left[0.886+4.64\left(\frac{a}{W}\right)-13.32{\left(\frac{a}{W}\right)}^{2}+14.72{\left(\frac{a}{W}\right)}^{3}-5.6{\left(\frac{a}{W}\right)}^{4}\right]\right\}}{\sqrt{{\left(1-\frac{a_{i}}{W}\right)}^{3}}}\right]$$

In Equations (1)–(4), the *pl* index denotes a plastic part and the *el* index denotes the elastic part. *υ* denotes the Poisson ratio, E is the elastic modulus and P stands for loading. The remaining symbols are related to specimen geometry. W is a dimension characteristic of the specimen, B_N_ is the specimen net width and a_i_ is the fracture current length. It needs to be noted that for the analyzed case, B_N_ is the same as B because the specimen used for the tests does not have side grooves. The remaining quantities are related directly to the fracture current length and are calculated using:(5)$$b=\left(a-W\right)$$
and
(6)$$\eta_{pl}=2+0.522\frac{b}{W}$$

In Equation (2), A_pl_ represents the static plastic field under the force–force application displacement point diagram, and this quantity was calculated using appropriate mathematical formulas included in additional dedicated standards [32].

AA2519 aluminum alloy was tested under two different temperature conditions. Therefore, two different elastic moduli, adequate for the test conditions, were used for fracture toughness calculations. Thus, E = 67.5 GPa was accepted for ambient conditions and, for cryogenic conditions Young’s modulus was accepted to be 80.3 GPa. Poisson’s ratio was accepted to be υ = 0.33 for both cases.

The maximum experimental values of the J–J_Q_ integral were calculated for each specimen and both temperature conditions. Averaged values of the obtained results are presented in Table 2. It needs to be noted that P_max_ is the maximum load used in the experiment and K_Pmax_ in the value of strain intensity calculated for the i-th point for a force equal to P_max_.

### 4.4. J–R Curve

To find an experimental value of the J–J_Q_ integral, diagrams of J–R curves were constructed with points marked on them according to the earlier presented mathematical dependencies. The diagram was constructed according to the directives presented in ASTM E 1820–18. Naturally, a construction line and exclusion line were marked on the diagram. Each value was determined based on the intersection points and adequately to the analyzed cases. Exemplary diagrams constructed for ambient and cryogenic conditions, including the construction and exclusion lines, are presented in Figure 6.

To check whether the results meet the research method assumptions, the following conditions were verified:(7)$$B_{min}>\frac{10J_{Q}}{\sigma_{Y}}$$
and:(8)$$b_{min}>\frac{10J_{Q}}{\sigma_{Y}}$$

It should be pointed out that the value of the yield point was determined experimentally while maintaining a constant temperature throughout the experiment. Following the calculations, the fulfillment of the condition of the specimen uncracked fragment length and thickness was confirmed. As it was for ambient conditions, B_min_ and b_min_ should be 2.31 mm and for cryogenic conditions, 1.5 mm. In the considered case, the specimen thickness was 10 mm, and the length of the uncracked specimen fragment was 22 mm.

An averaged maximal experimental value of the J–J_Q_ integral determined for heat-treated AA2519 aluminum alloy was 66.3 kJ/m^2^ for ambient conditions, while the same quantity for cryogenic conditions was 87.3 kJ/m^2^.

## 5. Analysis of Test Results

### 5.1. Analysis of the Results of Experimental Tests

Cryogenic conditions caused a change in the tested aluminum alloy mechanical properties. The maximal force value reached during the experiment was higher for cryogenic conditions. Naturally, the mean value of the strain intensity coefficient increased along with an increase in the maximum force because this parameter changes in direct proportion to the force, as seen in Equation (4). Cryogenic conditions have a significant influence on the shape of the J–R curve. For ambient conditions, the curve is more inclined to the x-axis, while for cryogenic conditions, the curve is more vertical.

Due to the phenomena described in the previous paragraph, in cryogenic conditions, the maximal experimental J–J_Q_ integral, determined by the R curve method, drops although the maximal force obtained in the tests increases. A rapid drop in the recorded force (in the experiment), indicating an increase in the fracture growth, occurs for the specimen with a smaller opening.

### 5.2. Microstructures of Specimen Fractures

SEM of the specimen fractures was performed for the parameters described in previous chapters. Three very clear zones can be observed for all specimens, both those tested in ambient temperature and in cryogenic temperature. The first (I) is the fatigue crack area, second (II) is the crack propagation area and the third (III) is the plastic hinge zone. Due to the research objective, zones (I) and (III) have not been described.

The transition between phase (I) and (II), recorded during SEM analysis of AA2519 aluminum alloy for both temperature conditions, is presented in Figure 7.

The fatigue fracture boundary is very clear in both cases. A wavy structure on the left side of both images shows gradual cracking under cyclic mechanical loads.

The shapes of fractures on the specimens tested under ambient conditions and under cryogenic conditions were diversified mainly due to topographic factors, but also in terms of crystallographic structures that occur on them.

Considering only zone (II), it must be stressed that the fracture that resulted directly from a crack was generally of plastic character (Figure 8). In ambient conditions, its entire surface was dominated by large dimples due to void connections caused by large undissolved particles of the θ phase. Not all the voids were connected to each other. This was manifested by the point occurrence of single voids all over the fracture surface. They were accompanied by medium-sized dimples connected with small particles. There were sheets of small voids between the primary dimples coming from the dispersion phase particles that grow and connect during primary void coalescence. Ductile fractures on the border of grains and between grains made up the smallest group, characterized by numerous, tiny dimples on the facets observed on the fracture [2]. The distribution of all types of dimples was similar all over the fracture.

In cryogenic conditions, the share of large primary dimples was bigger than that of intergranular fractures, especially in the initial phase of the fracture growth, that is, near the border of a fatigue crack. The length of the fracture growth was accompanied by an increase in the number of larger dimples. A decrease in the number of microfractures was observed on the entire surface of the fracture. This phenomenon can have an impact on the values of the experimental J–J_Q_ integral, force P_max_ and crack opening, because the occurrence of a larger number of microfractures under ambient conditions causes loss of material integrity, which leads to higher strain, and due to the cross-section reduction, the destructive force required is lower. It needs to be noted that the material tested under ambient conditions also exhibited a large number of microfractures in area (I) of Figure 7a.

Particles of undissolved phase θ phase seen in dimples lost their integrity under ambient conditions, that is, transverse fractures appeared in them. Generally, they were granulated.

The above conclusion is consistent with the conclusions included by Boroński et al. [18], where the static properties of AA2519 were presented.

## 6. Conclusions

This study presented the results determining the maximum experimental value of the J–J_Q_ integral characteristic to provide an assessment of the impact of temperatures down to 77 K on fracture toughness in terms of elasticity–plasticity. Experimentally generated fractures were analyzed using SEM. The following conclusions were formulated based on the test results.

(1)The maximum experimental value of the J–J_Q_ integral, determined using the J–R curve and a ratio of dimensions W/B = 4, met the requirements to be considered as a mechanical characteristic that could be used for comparing the material fracture toughness.(2)Cryogenic conditions had a significant influence on mechanical properties AA2519. In the case of the experimental value of the J–J_Q_ integral, cryogenic conditions caused an increase in the described value by an average of 31.6%, compared to ambient conditions.(3)A temperature of 77 K caused significant changes in the specimen fracture as well. Under cryogenic conditions, the number of microfractures decreased, and the number of intergranular fractures increased.(4)A significant drop in temperature involved a change in the character of the J–R curve. For cryogenic temperature, the J–R curve was more vertical, which means that fewer length growths were observed in fractures during successive load cycles.

## Figures and Tables

**Figure 1 materials-13-01555-f001:**
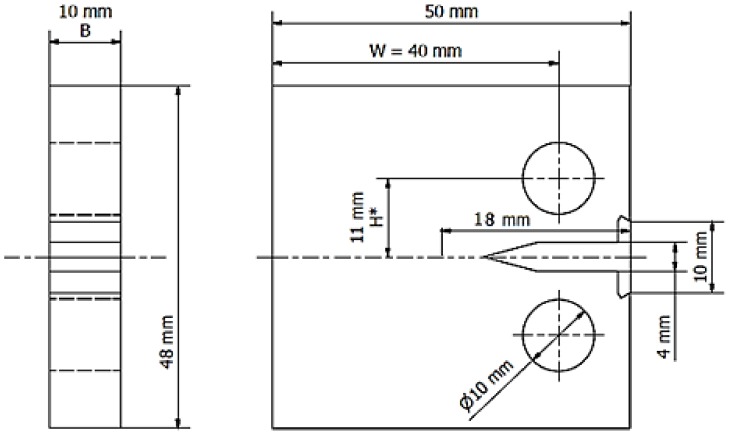
Basic dimensions of a compact specimen used for tests of fracture toughness.

**Figure 2 materials-13-01555-f002:**
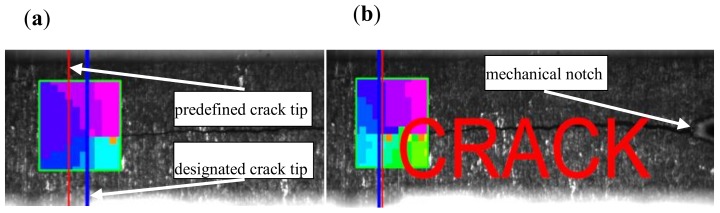
Scheme of the fracture peak identification, (**a**) top of the fracture before the borderline and (**b**) top of the fracture behind the borderline.

**Figure 3 materials-13-01555-f003:**
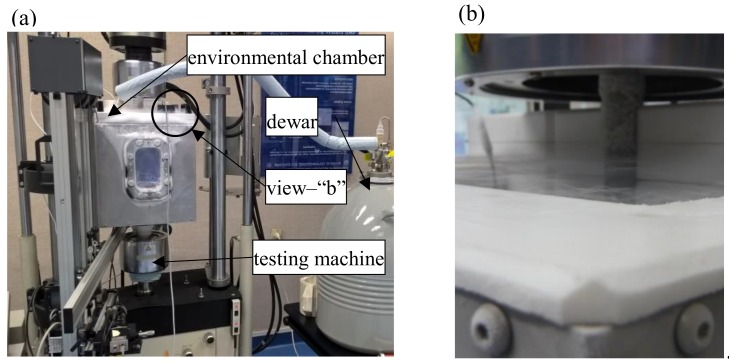
Environmental chamber used for tests at cryogenic temperature, (**a**) chamber mounted in a strength testing machine and (**b**) view of the machine inside.

**Figure 4 materials-13-01555-f004:**
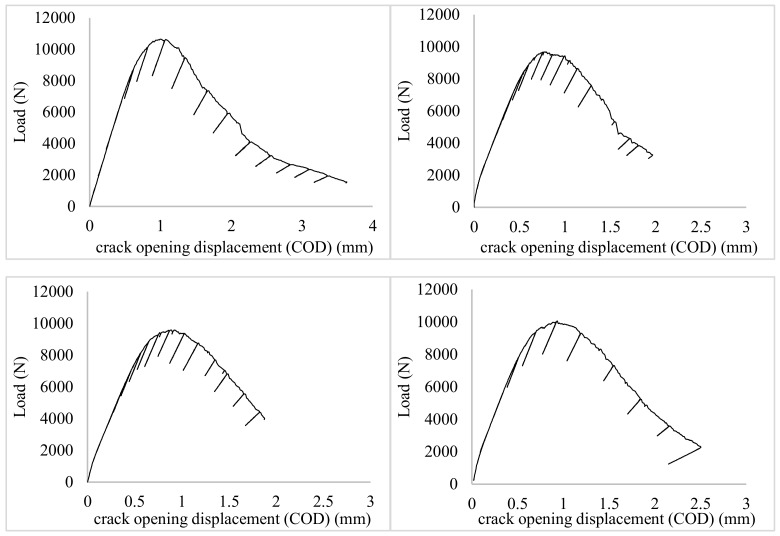
Force—crack opening displacement (COD) curve in compact tension specimens for AA2519 tested under ambient conditions.

**Figure 5 materials-13-01555-f005:**
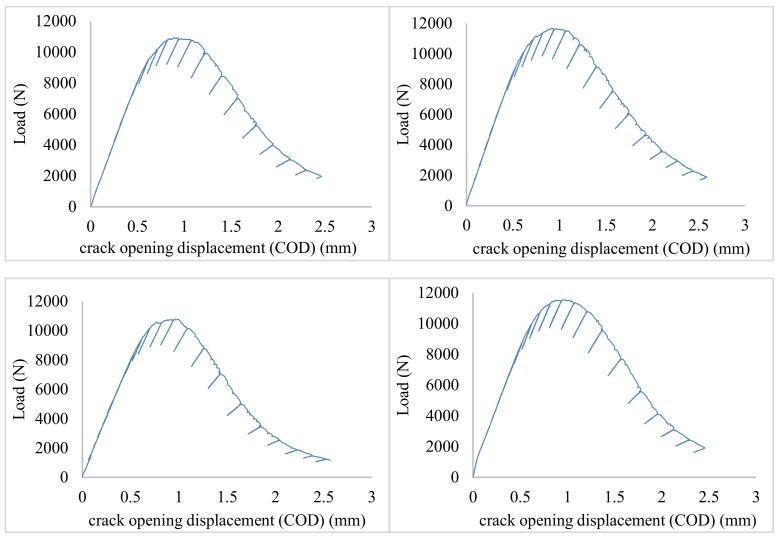
Force—COD curve in compact tension specimens for AA2519 tested under cryogenic conditions.

**Figure 6 materials-13-01555-f006:**
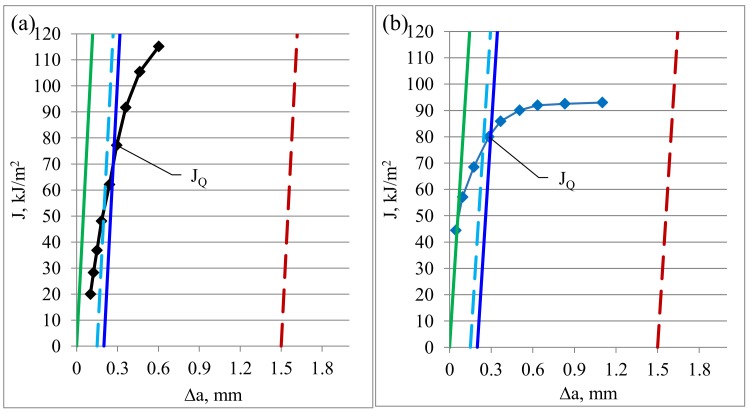
Shape of J–R curve for AA2519 aluminum alloy, (**a**) tested under ambient conditions and (**b**) tested under cryogenic conditions.

**Figure 7 materials-13-01555-f007:**
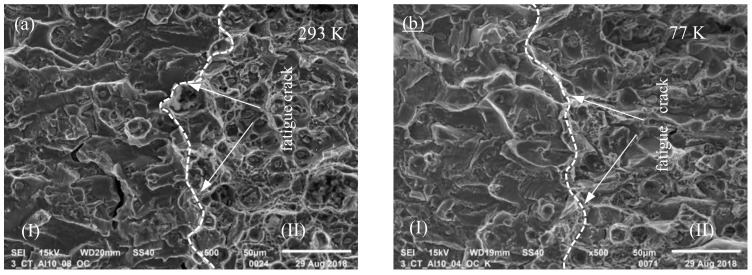
Fatigue crack boundary of compact tension specimens made from AA2519 aluminum alloy (**a**) tested in ambient conditions and (**b**) tested in cryogenic conditions.

**Figure 8 materials-13-01555-f008:**
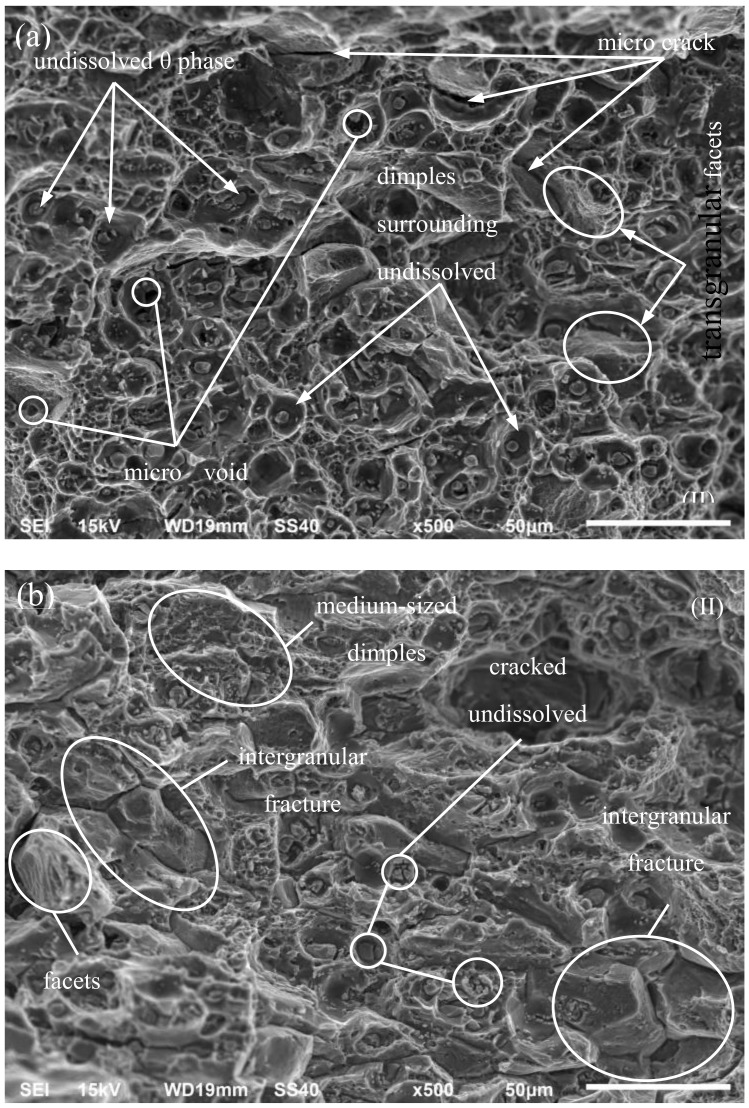
Image of the AA2519 (**a**) fracture obtained in tests performed at 293 K and (**b**) fracture generated in tests performed at 77 K.

**Table 1 materials-13-01555-t001:** The chemical composition of AA2519 aluminum alloy [9].

Si	Fe	Cu	Mg	Zn	Ti	V	Zr
0.06	0.08	5.77	0.18	0.01	0.04	0.12	0.2

**Table 2 materials-13-01555-t002:** Averaged tests results obtained by determination of the maximum experimental value of the J–J_Q_ integral.

Mechanical Characteristics	P_max_	K_Pmax_	J_Q_
	kN	MPa ∙ m^2^	kJ/m^2^
293 K			
average value	9950	47.7	66.3
standard deviation	528.3	2.7	7.6
77 K			
average value	11185	59.3	87.3
standard deviation	230	3.8	1.2
difference, %	ΔX = (X_77 K_ − X_293 K_)/X_293 K_∙100% where X-P_max_, K_Pmax_, J_Q_		
	12.4	24.3	31.6

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
