# Peer review of "The Influence of Cryogenic Conditions on the Process of AA2519 Aluminum Alloy Cracking"

_materials, 2020, doi:10.3390/ma13071555_

Round 1
Reviewer 1 Report
Introduction needs to be reorganized, because in its present state do not contain up to date literature review and do not explain why this research is necessary. I would advise to merge it somehow with the second chapter.
- L 92 - 113: Sample description is not clear. Authors stated that 8 mm fatigue fracture was generated, or 18 mm with mechanical notch. However, in the Fig. 1 is the length of the notch 20.5 mm?! Please explain that.
- L 103: “…the final thickness of the specimens, 10 mm, could lead to ambiguity.”
What is the purpose of the study of which is know that it could lead to ambiguity? Authors have to provide solid evidence that their results are not vague, for instance by extending of the number of tested samples or performing additional measurements of different specimens’ thicknesses.
- L 129, Fig. 2a: This figure seems to be damaged, there is no visible text in the text box. Where the crack is exactly located? It is not clear from the Fig. 2b. I advise to supplement these figures with macro photograph of the sample where the crack will be clearly visible.
- L 132: Authors mention “static properties”. What does static properties mean?
- L 153: “The device worked at 15 kV voltage.” What device? Authors should properly name all utilized instruments.
- Fig. 4 and 5: The load – COD curves for samples tested under room temperature seems to be far more scattered (less consistent) than results obtained in LN2 temperature. Please explain that.
- Tab. 2: To the averaged values of the obtained results authors should calculate also the standard deviation from the obtained data.
Author Response
Thank you very much for your suggestions and comments. I did my best to fill in all the gaps and clarify the doubts. Details of changes are sent in the attached file.

Reviewer 2 Report
The work needs some chagnes to make it readable and acceptable for publication. The authors have not mentioned the details of their work in an appropriate manner. Comments below.
-The introduction is not appropriate at all and needs major work before acceptance
-Line 34 - while maintaining plasticity .... what do the authors mean by that ?
-3 line paragraphs are NOT considered as paragraphs.
-section 2.1 contains many of the items that should be included in introduction
-how was heat treatment performed ? which furnace ? heating rate ? cooling rate ? holding time ? type of furnace ?
-How was strain measured ? was a DIC used ?
-Fig 2 is not clear and confusing. please make the figure clear and explain the details
-3.3 - what equipment was used ? where did you get the liquid N2 ? what was the purity of the liquid etc etc.
-line 152 - what tests ? what type of work ?
-How as the crack opening displacement measured ?
-Fig4 - what is the difference between the graphs ? Provide cyclic infromatin also please. Also stress-strain curves. They should be available based on the DIC data.
-CT is used for computed tomography . please use some other name. this is not appropriate name.
-Fig7 fatigue cracks are not visible. The reviewer is not conviced based on the figure. Please provide proper evidence.
-How was the theta identified ?i mean the theta phase ? No proper data has been provided.
-The authors have performed 2 tests under ambient and cryogenic conditions along with some fracture surfaces. There is no microstructure study (even though there is a metalographical section) and nothing else in the paper.
-The authors have claimed that this work is new. However, several researchers have looked into the cryogenic conditinos of otheraluminum alloys. so while the work might be new for AA2519, its not new in its concept. Hatamelah et al, Potti et al, Boronski et al (fracture), Rao et al (fracture, Anderson et al (ALSO WORKED on 2519), and Kotyk et al. (2519, fracture at cryogenic conditions)
-The title includes "alloy cracking". However, the cracking behivor is just shown with no major results.
The paper is similar to one by the same authors "Influence of the theoretical load point on the value of the J - JQ integral during determination of fracture toughness of 2519 aluminium alloy". Please explain the differences of the two works.
Author Response

(The authors gave the same response as above.)

Reviewer 3 Report
Take care of all mandatory changes in attached file.

Author Response

(The authors gave the same response as above.)

Round 2
Reviewer 1 Report
Authors definitely improved this manuscript. Therefore, now I can recommend to publish it in MDPI Materials.
Reviewer 3 Report
Good revision.
- Ref 7 is not aligned with the main ideas. Please is much better to define the difference found by Argentinian people in fatigue testpieces, in Alternatives for specimen manufacturing in tensile testing of steel plates, Experimental Techniques 40 (6), 1555-1565- Include this in final version.